# Superoxide dismutases maintain niche homeostasis in stem cell populations

Olivia Majhi, Aishwarya Chhatre[†], Tanvi Chaudhary, Devanjan Sinha*

Department of Zoology, Institute of Science, Banaras Hindu University, Varanasi, India

## eLife Assessment

In this work, the authors intend to assess the existence of a redox potential across germline stem cells and neighbouring somatic stem cells in the Drosophila testis. Some aspects of the manuscript are **convincing**, like the clear effect of SOD KD on cyst cell differentiation state. Other conclusions of the work, such as the non-autonomous effect of this KD on germ cells are not sufficiently supported by the data. This remains true even with the revised version of the paper, as the effect of redox state of the soma on the germline is a major point of the paper, and this remains a critical flaw. The work could be potentially **useful** if the critiques of the reviewers were fully addressed; the strength of the evidence of the manuscript as it stands is still **inadequate**. Readers should use their own judgment about the validity and meaningfulness of different findings.

*For correspondence:
devanjan@bhu.ac.in

Present address: [†]Biological Research Centre, Institute of Genetics, Eötvös Loránd Research Network (ELKH), Szeged, Hungary

Competing interest: The authors declare that no competing interests exist.

**Abstract** Reactive oxygen species (ROS), predominantly derived from mitochondrial respiratory complexes, have emerged as key molecules influencing cell fate decisions like maintenance and differentiation. These redox-dependent events are mainly considered to be cell intrinsic in nature; on the contrary, our observations indicate involvement of these oxygen-derived entities as intercellular communicating agents. In *Drosophila* male germline, Germline Stem Cells (GSCs) and neighbouring Cyst Stem Cells (CySCs) maintain differential redox thresholds where CySCs have higher redox state compared to the adjacent GSCs. Disruption of the redox equilibrium between the two adjoining stem cell populations by depleting Superoxide Dismutases (SODs), especially Sod1, results in deregulated niche architecture and loss of GSCs, which was mainly attributed to loss of contact-based receptions and uncontrolled CySC proliferation due to ROS-mediated activation of self-renewing signals. Our observations hint towards the crucial role of differential redox states where CySCs containing higher ROS function not only as a source of their own maintenance cues but also serve as non-autonomous redox moderators of GSCs. Our findings underscore the complexity of niche homeostasis and predicate the importance of intercellular redox communication in understanding stem cell microenvironments.

## Introduction

Studies from the past few decades have shown the apparent role of ROS in influencing various biological processes (*Schieber and Chandel, 2014*; *Holmström and Finkel, 2014*; *D'Autréaux and Toledano, 2007*). ROS are usually produced in specific cellular compartments close to their target molecules to regulate important signaling pathways (*Maryanovich and Gross, 2013*). Mitochondria, a major source of oxidant species (*Murphy, 2009*; *Hamanaka and Chandel, 2010*), localize dynamically towards the nucleus, effecting oxidation-induced reshaping of gene expression profiles (*Al-Mehdi et al., 2012*). Localised ROS production are contributed by NADH oxidases distributed across different cellular regions (*Bedard and Krause, 2007*). Intracellular hydrogen peroxide gradients, maintained by the thioredoxin system, allow intercompartmental exchanges among endoplasmic reticulum (ER),

mitochondria, and peroxisomes (*Appenzeller-Herzog et al., 2016*; *Mishina et al., 2019*; *Yoboue et al., 2018*).

The process of redox relays is conserved and plays a fundamental role in self-renewal and differentiation of stem cell populations (*Rehman, 2010*). Stem cells, whether embryonic or adult, generally maintain a low redox profile, barring a few exceptions, and are characterised by subdued mitochondrial respiration (*Li and Marbán, 2010*; *Le Belle et al., 2011*; *Chuikov et al., 2010*; *Owusu-Ansah and Banerjee, 2009*; *Tan et al., 2017*). Levels of ROS are tightly regulated, and elevated amounts promote early differentiation and atypical stem cell behaviour (*Zhou et al., 2016*). However, leading evidence suggests that many transcription factors require oxidative environments for the maintenance of pluripotent states. For instance, physiological ROS levels play a crucial role in genome maintenance of embryonic stem cells (*Li and Marbán, 2010*). Multipotent hematopoietic progenitors, intestinal stem cells (*Morris and Jasper, 2021*), and neural stem cells require relatively high baseline redox for their maintenance (*Le Belle et al., 2011*; *Owusu-Ansah and Banerjee, 2009*; *Sinenko et al., 2011*). Suppression of the Nox system or mitochondrial ROS compromises the maintenance of these self-renewing populations and promotes their differentiation or death (*Zhou et al., 2016*). These evidences point towards the essentiality of a well-tuned redox state for balancing the pluripotent and differentiated states. However, how stem cells regulate their redox potential by possessing a restrained oxidant system is not very clear.

We addressed this fundamental question in two-stem cell population-based niche architecture in *Drosophila* testis. The testicular stem cell niche is composed of a central cluster of somatic cells called the hub which contacts eight to eleven GSCs arranged in a round array (*Davies and Fuller, 2008*; *Hime et al., 1996*). A pair of cyst stem cells (CySCs) enclose each GSC and make their independent connections with the hub and GSCs via adherens junctions (*Gönczy and DiNardo, 1996*; *Fabrizio et al., 2003*; *Hardy et al., 1979*; *Chen et al., 2013*; *Inaba et al., 2010*; *Greenspan et al., 2015*). The hub cells secrete self-renewal factors essential for both GSCs and CySCs maintenance involving signalling cascades like Jak-Stat signalling (*Kiger et al., 2001*; *Tulina and Matunis, 2001*), BMP (Bone Morphogenetic Signalling) (*Kawase et al., 2004*), Hedgehog signalling (*Michel et al., 2012*). The hub and CySCs produce BMPs which repress GSC differentiation by suppressing transcription of bag-of-marbles (Bam) (*Kawase et al., 2004*; *Song et al., 2004*). Asymmetric division of both GSC and CySC is essential for proper cyst formation (*Losick et al., 2011*; *Lenhart and DiNardo, 2015*). A developing cyst contains a dividing and differentiating gonialblast encircled by cyst cells that provide nourishment to developing spermatids (*Jemc, 2011*; *Zoller and Schulz, 2012*). However, the maintenance of GSCs in the spatially controlled microenvironment is still not very clear.

We found the existence of a balanced differential redox state between GSC and CySC to be essential for niche homeostasis. CySCs, by virtue of their higher redox threshold and more clustered mitochondria generated an intercellular redox state that affected the physiological ROS levels in GSCs. Alterations in CySC redox state affected the self-renewal and differentiation propensities of the GSCs. Intercellular redox imbalance disrupts niche homeostasis by inducing premature differentiation of germline stem cells (GSCs) and aberrant proliferation of CySCs, driven by activation of pro-proliferative signaling pathways and attenuation of cell-cell contact-mediated communication. Our results indicated a sophisticated interplay between these two stem cell populations, where a higher redox state in CySCs not only supports their self-renewing processes but also non-autonomously promotes the maintenance of GSCs.

## Results

### CySCs maintain higher differential ROS in comparison to adjacent GSCs

The apical region of the *Drosophila* testis incorporates a cluster of differentiated cells, the hub, which is surrounded by GSCs in a rosette arrangement with each GSC being enclosed by two CySCs (*Figure 1A*). Immunostaining the ATP5A subunit of mitochondrial ATPase and marking cell boundary by discs-large (Dlg) indicated different mitochondrial distribution among these two stem cell populations. In wild-type adult testis, we observed sparsely populated mitochondria in Vasa[+] GSCs as compared to their dense distribution in CySCs (*Figure 1B#, C, D*, and *Figure 1—figure supplement 1B–B"*). The number of mitochondria per cell was higher in CySC (*Figure 1F*). The ATP5A labelling overlapped with *TFAM-GFP*, a mitochondrial transcription factor (*Larsson et al., 1998*) that labelled

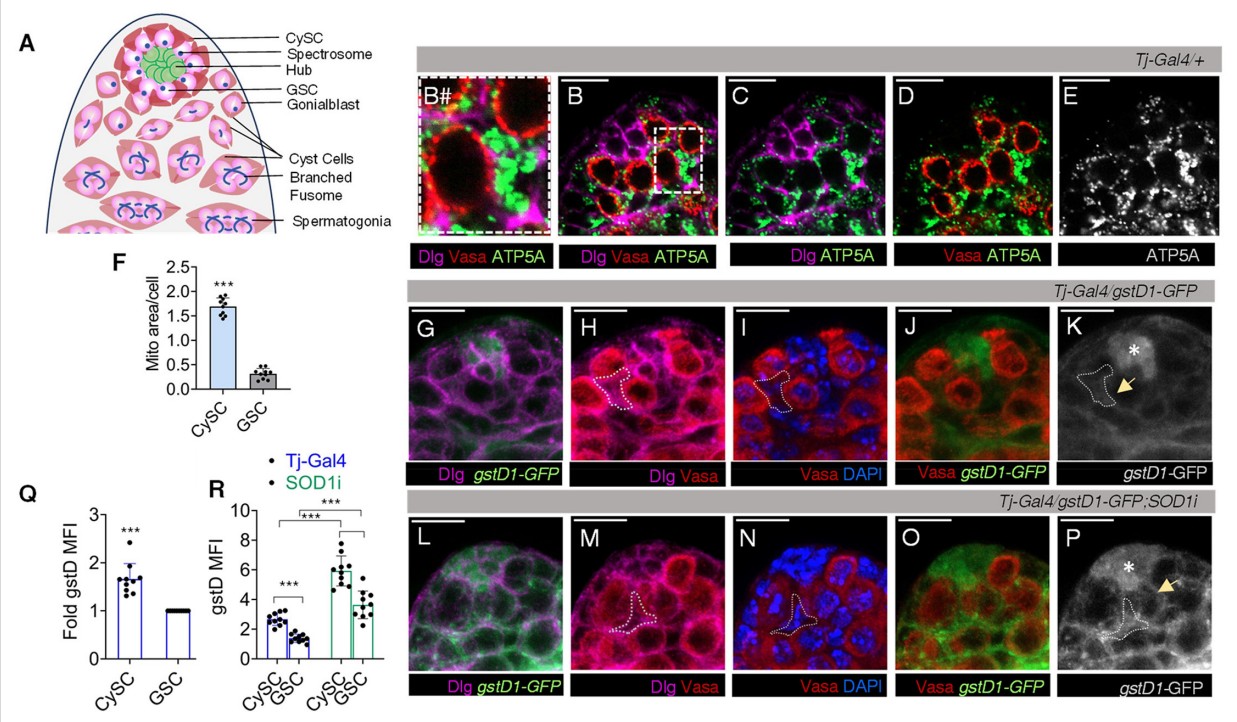

**Figure 1.** Mitochondrial distribution and redox profile of GSCs and CySCs in adult *Drosophila* testes. (**A**) Schematic representation of adult *Drosophila* testicular niche showing arrangement of different stem cell populations (GSC-Germinal stem cell, CySC-cyst stem cell). (**B–E**) Differential distribution of mitochondria labelled with ATP5A (monochrome/green) in Vasa+ GSCs of wild-type fly testis with cellular boundaries marked by Dlg; B# shows a digitally zoomed image of the dotted area of B. (**F**) Quantification of mitochondrial area per cell. Bars represent mean ± s.e.m., n=10 fields of view; ***p (unpaired t-test)<0.0001. (**G–P**) Redox profiling of testicular stem cell niche using *gstD1*-GFP as intrinsic reactive oxygen species (ROS) reporter in control (**G–K**) and *Tj-Gal4* driven *Sod1RNAi (Sod1i)* testis (**L–P**). Dotted area denotes the region of CySC occupancy and asterisk denotes the hub. (**Q**) Quantification of *gstD1*-GFP mean fluorescence intensity (MFI) represented as fold change between GSCs and CySCs in controls. Data denotes mean ± s.e.m., n=25, ***p (unpaired t-test)<0.0001. (**R**) Quantification of cell-specific ROS content upon *Sod1RNAi*. Data denotes mean ± s.e.m., n=25, ***p (punpaired t-test)<0.0001. Controls are the indicated driver line crossed with *Oregon R+*. Scale bar: 10 μm.

The online version of this article includes the following figure supplement(s) for figure 1:

**Figure supplement 1.** Mitochondrial distribution and effect of elevated cyst stem cell (CySC) reactive oxygen species (ROS) on stem cell numbers.

the mitochondria, further confirming the patterning observed between these two stem cell populations (*Figure 1—figure supplement 1A*). Since mitochondria are known to be one of the major producers of ROS in cells, we tested for the relative redox profiles in these two stem cell populations. Mitochondrial dispersion pattern in GSCs and CySCs corresponded with the intensity variance of the ROS reporter line *gstD1*-GFP (*Sykiotis and Bohmann, 2008*) among the different cell populations at the niche. CySCs (outlined by a dotted boundary) exhibited distinctly higher *gstD1* reporter intensity than the neighbouring GSC (shown with yellow arrowhead) (*Figure 1K*, *Figure 1—figure supplement 1C‴*). Quantification of this intensity difference indicated that CySCs exhibited a higher baseline level of ROS compared to GSCs (*Figure 1Q*). The hub zone (denoted by asterisk) also presented higher *gstD1*-GFP intensity when compared to the surrounding germline (*Figure 1K*, *Figure 1—figure supplement 1C‴*).

Given the ability of free radicals to diffuse, we hypothesised that the somatically derived cells in the niche, by virtue of their higher ROS state, might be maintaining the redox environment of their surroundings. In this study, we evaluated the redox interplay between the two stem cell populations in the niche. To test the role of CySC in influencing GSC redox state, we asked if disrupting the superoxide dismutases in CySCs would influence the redox state of GSCs. To deplete SOD1, we used *Tj-Gal4* (*Fairchild et al., 2016*) driver (*Tj >Sod1* i) that expresses majorly in CySCs and early differentiating cyst cells (CCs) with minor levels of expression in the hub. Depletion of SOD1 majorly in CySCs resulted in a net increase in *gstD1*-GFP intensity in the niche zone (*Figure 1P, R*, *Figure 1—figure supplement 1E*), and an overall rise in superoxide levels, confirmed through DHE fluorescence

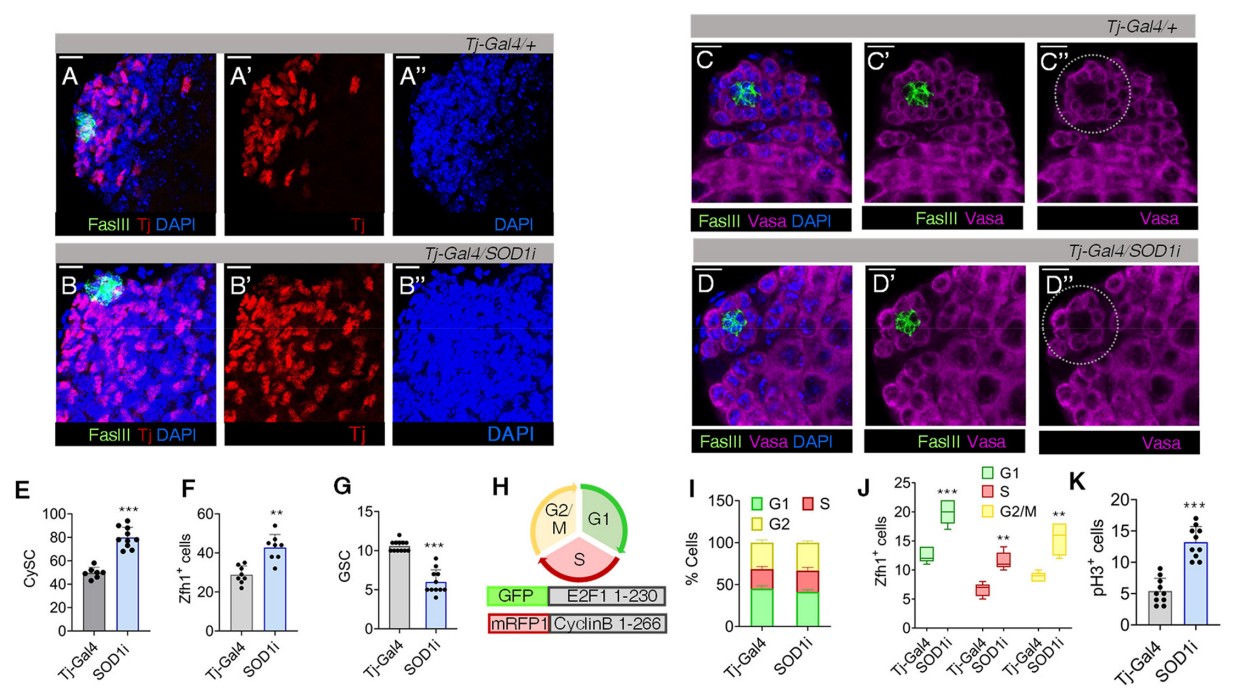

**Figure 2.** Redox disequilibrium in the cyst stem cell lineage deregulates early cyst stem cells (CySCs) and decreases the germline stem cell (GSC) number. (**A–B**) Distribution of Tj⁺ CySC/early cyst cell around the hub (FasIII) in control and Tj driven *Sod1RNAi* testis. (**C–D**) Represents the rosette arrangement of Vasa⁺ GSCs flanking the hub (dotted area). (**E–G**) Mean number of Tj⁺ cells, Zfh1⁺ early CySCs and GSCs in control and *Sod1RNAi* lines shown as mean ± s.e.m, n=10, ****p (unpaired t-test)<0.0001, ***p (unpaired t-test)<0.0001. (**H**) Construct the design of the fluorescent ubiquitination-based cell cycle indicator (FUCCI) reporter line for tracking the cell cycle stages in *Drosophila* tissues, containing degrons of Cyclin B and E2F1 proteins fused with RFP or GFP. G1, S, and G2/M is represented by GFP⁺, RFP⁺, and dual labelled cells, respectively. (**I–J**) Comparative changes in the cell cycle phases (**I**) or Zfh1⁺ cells (**J**) present in G1, S, and G2/M phase at the niche zone between control and *Tj >Sod1* i. Data denote mean ± s.e.m, n=10, ****p (unpaired t-test)<0.0001, ***p (unpaired t-test)<0.001. (**K**) Mean number of pH3 + cells, a mitotic marker. Data points denote mean ± s.e.m, n=10. Scale bar: 10 μm.

The online version of this article includes the following source data and figure supplement(s) for figure 2:

**Figure supplement 1.** Niche composition upon depletion of Sod paralogs in cyst stem cells (CySCs) or germline stem cells (GSCs).

**Figure supplement 1—source data 1.** Individual original raw unlabelled blots to *Figure 2—figure supplement 1N*.

**Figure supplement 1—source data 2.** Labelled original blots corresponding to *Figure 2—figure supplement 1N*.

analysis (*Figure 1—figure supplement 1F*). Increased *gstD1* labelling due to SOD1 depletion in CySC (denoted by dotted area) (*Figure 1P*, *Figure 1—figure supplement 1D'''*) adjacent to Vasa⁺ GSC (shown with yellow arrow) resulted in concomitant increase in *gstD1* intensity at the Vasa⁺ GSC (*Figure 1P*, *Figure 1—figure supplement 1D'''*). This indicates that the elevated redox state in CySCs led to a corresponding increase in ROS levels in adjacent GSCs (*Figure 1R*), suggesting a potential redox crosstalk between these stem cell populations, where the redox state of CySCs might be influencing the oxidative status of neighbouring GSCs. This crosstalk was further validated by subsequent phenotypic analysis.

## Balanced CySC redox profile is crucial for its proliferation and GSC maintenance

Alongside the differential redox profile, we observed that elevated ROS levels in *Tj >Sod1* i had a striking effect on the increase of DAPI⁺ nuclei at the testis tip (*Figure 2A''*). This overcrowding was attributed to both an increase in the number of Tj⁺ CySCs and early differentiating CCs, as well as their positional shift (*Figure 2A', B' and E*). To further validate these findings, we used an alternative driver line, *C-587-Gal4*, which specifically drives expression in CySCs and CCs. This also resulted in an increase in the total number of Tj⁺ cells (*Figure 1—figure supplement 1H' and M*), although to a lesser extent than *Tj-Gal4*, suggesting that signals from the hub may also partially influence CySC

proliferation. Given the stronger phenotypic effect observed with *Tj-Gal4*, we proceeded with this driver line for subsequent experiments.

The enhancement in the number of CySCs was also reflected by a ~ twofold change in Zfh1+ cells (*Figure 2F*, *Figure 2—figure supplement 1R(I)-S(I)*). However, Vasa+ cells flanking the hub (GSCs) showed a substantial reduction in number (*Figure 2C"–D" and G*). This decline was further validated by western blot analysis, which revealed lower overall detectable levels of Vasa protein, indicating a potential impact of the altered redox state on germline maintenance (*Figure 2—figure supplement 1N*). Since Tj shows a minor expression in hub cells, we validated our phenotypic data using *C-587-Gal4* that is specific for CySCs and CCs. Similar to *Tj >Sod1*, *C-587-Gal4>Sod1* i also showed a reduction in the number of neighbouring GSCs (*Figure 2—figure supplement 1N*). The observations were further verified using *Sod1i* localised in different chromosomes to avoid any chromosome or balancer-based bias and a similar result was obtained where alterations in CySC ROS affected GSC number (*Figure 2—figure supplement 1I–L and O-P*). However, the phenotypic effect of higher ROS was limited to its origin in CySCs only because ablation of GSC redox status did not cause any marked change in niche composition. *Nos-Gal4* driven knock-down of Sod1 in GSCs did not result in a significant change in CySC number (*Figure 2—figure supplement 1A', B' and E*) but effected a considerable reduction in Vasa+ cells (*Figure 2—figure supplement 1C', D' and F*), aligning with previous observations (*Tan et al., 2017*). Although Sod1 is the predominant enzyme which also localizes in the intermembrane space of mitochondria, we observed a similar result upon depleting the matrix-localised Sod2 in both GSCs and CySCs (*Figure 2—figure supplement 1G–M*), indicating the involvement of mitochondrial ROS for the observed cellular phenotypes.

To confirm the active proliferation of cells and rule out the possibility of arrested growth, we utilised the fly-Fluorescent Ubiquitination-based Cell Cycle Indicator (FUCCI) system (*Zielke et al., 2014*) which comprises two reporter constructs marking G1/S transition (green), S (red), and G2/early mitosis (yellow) (*Figure 2H*). Testes from *Tj >Sod1* i exhibited an approximately twofold increase in cell number across all stages of the cell cycle (*Figure 2—figure supplement 1O*). However, the percentage of cells in each phase remained unchanged (*Figure 2I*), indicating that the observed increase in different phases is a consequence of more cells entering proliferation rather than alterations in the duration of different cell-cycle stages. The enhanced number of dividing cells was contributed mainly by progressive division of CySCs, showing > twofold difference in the accumulation of Zfh1+ nuclei in S and G2/M phases (*Figure 2J*, *Figure 2—figure supplement 1RV-VII and SV-VII*). The number of Zfh1-labelled cells in G1/S transition was also substantially more (*Figure 2—figure supplement 1RIII and SIII*), indicating dysregulated redox ensued an increased mitotic index of CySCs. The enhanced mitotic activity of these proliferating niche cells is further supported by increased phospho-histone H3 (PH3) incorporation, indicating a higher mitotic frequency compared to the control (*Figure 2K*, *Figure 2—figure supplement 1T-U*), along with elevated cyclin D expression (*Figure 2—figure supplement 1Q*). Downregulation of Sod1 in GSCs through *Nos-Gal4* demonstrated no significant change in G1/S, S, and G2/M phase numbers, confirming our previous observation that altering ROS levels in GSCs does not have any non-autonomous effect on CySC numbers (*Figure 2—figure supplement 1P*).

## CySC-induced ROS couples precocious differentiation of stem cell lineages

Since in *Tj >Sod1* i testes, the increased number of Tj+ cells, marking CySCs and early differentiating cells, exceeded the number of CySC-specific Zfh1+ cells (*Figure 2J*, *Figure 2—figure supplement 1O*), we checked for parallel enhancement of cellular differentiation using the corresponding marker, *Eya* (Eyes absent).

We found that Eya+ cells clustered towards the hub, along with a significant increase in their number in Sod1-depleted CySCs (*Figure 3A, B and D*). This increased number of Eya+ cells does not suggest that these differentiating cells are proliferative. Instead, we propose that the knockdown of Sod1 may alter the timing or regulation of cyst cell differentiation, leading to an accumulation of Eya+ cells near the niche. The supposed premature expression of Eya resulted in a population of differentiating cyst precursor cells co-expressing Eya-Zfh1 (*Figure 3A", B" and D*). This population was found to be deviating from the normal partitioning of wild-type representative populations (*Figure 3C*). In contrast, depletion of *Sod1i* in the germline lineage using *Nos-Gal4* did not result in a similar phenotypic arrangement (*Figure 3—figure supplement 1A–D*). These findings suggest that

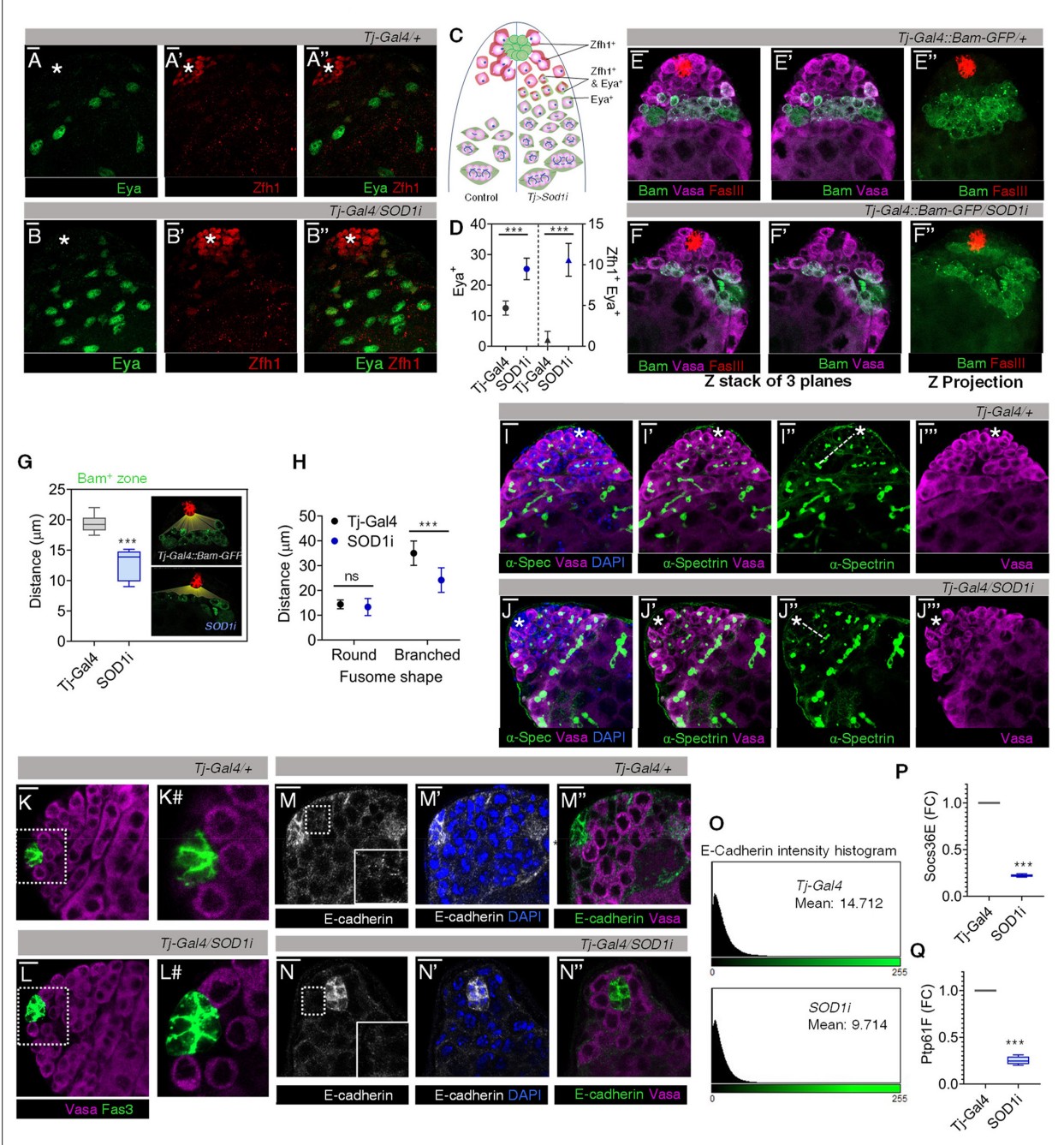

**Figure 3.** Reactive oxygen species (ROS) imbalance in cyst stem cells (CySCs) promotes differentiation of both germline stem cells (GSCs) and CySCs. (**A–D**) Number of early CySCs (Zfh1+) (**A′- B′**), late differentiating CCs (Eya+) (A - B) and Zfh1, Eya co-expressing population (**A″- B″**) were imaged, represented schematically (**C**), and quantified with respect to control and *Sod1RNAi* lines (**D**). Data points represent mean ± s.e.m, n=10, ****p (unpaired t-test)<0.0001*. (**E–F**) The effect of Sod1 depletion in CySCs on the differentiation status of GSCs as observed using *Bam-GFP* reporter line. (**G**) The relative distance of the differentiation initiation zone (Bam+) from the hub (red) in control and *Sod1i* testis quantified in single sections and shown as mean ± s.e.m, n=10, ****p (unpaired t-test)<0.0001*. (**H–J**) The shape and size of the spectrosomes marked with α-spectrin (green) were imaged (**I″-J″**) and quantified for their distance from the hub (marked with asterisk) (**H**), n=10, ****p (unpaired t-test)<0.0001*. Branched fusome marks differentiating populations (**I′-J′**). (**K–L**) Represents the rosette arrangement of Vasa+ GSCs flanking the hub. The digitally magnified region around the hub (dotted line) as seen in control (K#) and *Sod1RNAi* (L#). (**M–N**) Comparative staining of CySC-GSC contacts through adherens junction using E-cadherin (monochrome/green). Dotted area near the hub has been expanded as an inset to show loss of E-cadherin network. (**O**) E-cadherin intensity histogram plot generated from ImageJ representing the mean intensity of expression in the region flanking the hub and GSCs. (**P–Q**) Fold change in expression of Stat-dependent transcripts Socs36E (**P**) and Ptp61F (**Q**) among control and *Sod1i* niche, obtained through qPCR. Data is shown as mean ± s.e.m, n=3, ****p (unpaired t-test)<0.0001*. ns: not significant. Scale bar: 10 µm.

*Figure 3 continued on next page*

*Figure 3 continued*

The online version of this article includes the following figure supplement(s) for figure 3:

**Figure supplement 1.** Enhanced reactive oxygen species (ROS) in germline stem cells (GSCs) has no effect on early CySCs and late cyst cells, but changes in CySCs affect differentiation and cell-cell adhesion.

alterations in the redox state of GSCs can impact their numbers, even when the changes are induced non-autonomously through CySCs (*Tan et al., 2017*).

Together with the reduction in Vasa⁺ GSCs shown earlier (*Figure 2G*), we also detected gonialblasts expressing the differentiation-promoting factor Bam to be much closer to the hub in *Tj>Sod1* i testes (*Figure 3E" and F"*). The mean distance between the hub and Bam⁺ cells was found to be shortened by ~7 microns (*Figure 3G*). The relative advancement of spermatogonial differentiation towards the hub was also ascertained by tracking the transformation of GSC-specific spectrosome into branched fusomes connecting the multi-cell gonialblast (*Figure 1A*). The branching fusomes were found more proximal to the hub than controls (*Figure 3H, I" and J"*), along with parallel reduction in the number of spectrosomes, implying a decline in early-stage germ cells (*Figure 3—figure supplement 1F*). This loss could possibly be attributed to loss of intercellular contacts, particularly with hub cells (*Figure 3K# and L#*). The observation corroborated with decreased expression pattern of E-cadherin flanking the hub and GSCs, in *Tj>Sod1* i (*Figure 3M–O*) and could be one of the reasons for GSCs dissociation and differentiation (*Figure 3L#*). Disengagement of GSCs from the hub caused reduction in Socs36E and Ptp61F transcripts, which subsequently altered Stat expression, a key driver of E-cadherin expression (*Figure 3P and Q*). These results suggest that the disruption of intercellular redox gradient affected the premature differentiation of stem cells in the niche.

## Disrupted niche architecture compromises GSC-CySC communication

GSCs and CySCs intercommunicate through several factors, such as EGFR, PI3K/Tor, Notch (*Ng et al., 2019*) to support CySC maintenance (*Amoyel et al., 2016b*). EGF ligands secreted by spermatogonia maintain differences in cell polarity and segregate self-renewing from differentiating populations (*Castanieto et al., 2014*). *Tj >Sod1* i cells showed lower levels of cell-polarity marker Dlg (*Figure 4A", B" and E*), that expanded more into the differentiated zone (*Figure 4—figure supplement 1A*), indicating loss of cell polarity and aligning with the overtly proliferative nature of these cells. Along with Dlg, a net reduction in pErk levels was observed across sections in *Sod1i* testis (*Figure 4C', D' and F*). This reduction in overall Erk levels may be attributed to the loss of cell contact and altered GSC-CySC balance in *Sod1i* testis. The proliferation of cyst cells correlated with increased levels of self-renewal inducers, including the elevated expression of Hedgehog (Hh) pathway components (*Michel et al., 2012*), in *Tj >Sod1* i testis. The depletion of Sod1 in CySC resulted in elevated expression of Hh receptor, Patched (Ptc) (*Figure 4I*, *Figure 4—figure supplement 1V*), which extended to regions farther from the hub, overlapping with expanded Tj⁺ cells (*Figure 4G and H*). Since Ptc itself is a transcriptional Hh target, its expression in CySCs suggests active Hh signalling (*Chen and Struhl, 1996*), further supported by higher levels and a similar pattern of Hh transcriptional effector, Cubitus interruptus (Ci), in *Sod1i* testis (*Figure 4M", N", J*, *Figure 4—figure supplement 1W*). Since Ptc, Ci, and a GPCR-like signal transducer Smoothened (Smo) are all transcriptionally driven by Hh, and Hh itself is regulated by higher levels of Ci (*Chen and Struhl, 1998*; *Jiang and Hui, 2008*), we expectedly found higher levels of transcripts for these pathway genes (*Figure 4—figure supplement 1C–F*). The overt proliferation of CySCs under high ROS conditions was suppressed by reducing hh mRNA levels in hub cells and CySCs using *Hh-RNAi* (*Figure 4L*, *Figure 4—figure supplement 1G-I*), which also rescued the GSC population (*Figure 4K*, *Figure 4—figure supplement 1J-L*). The levels of Sod1 transcripts under depletion of Sod1 alone and Sod1, hh double knock-down were almost similar (*Figure 4—figure supplement 1M*). To determine whether the changes observed in the germ cell population were a sole consequence of *Sod1* knockdown in CySC, we manipulated CySC numbers using *UAS-Ci* and assessed their impact on neighbouring GSCs. Ci overexpression driven by *Tj-Gal4* led to a higher number of Tj⁺ cells (*Figure 4—figure supplement 1N, R" and S-U*) and a complete loss of Vasa⁺ GSCs (*Figure 4—figure supplement 1O and R'''*). However, the effect on CySC and GSC population was less severe when driven by *C-587 Gal4* (*Figure 4—figure supplement 1N, O, Q"and Q'''*).

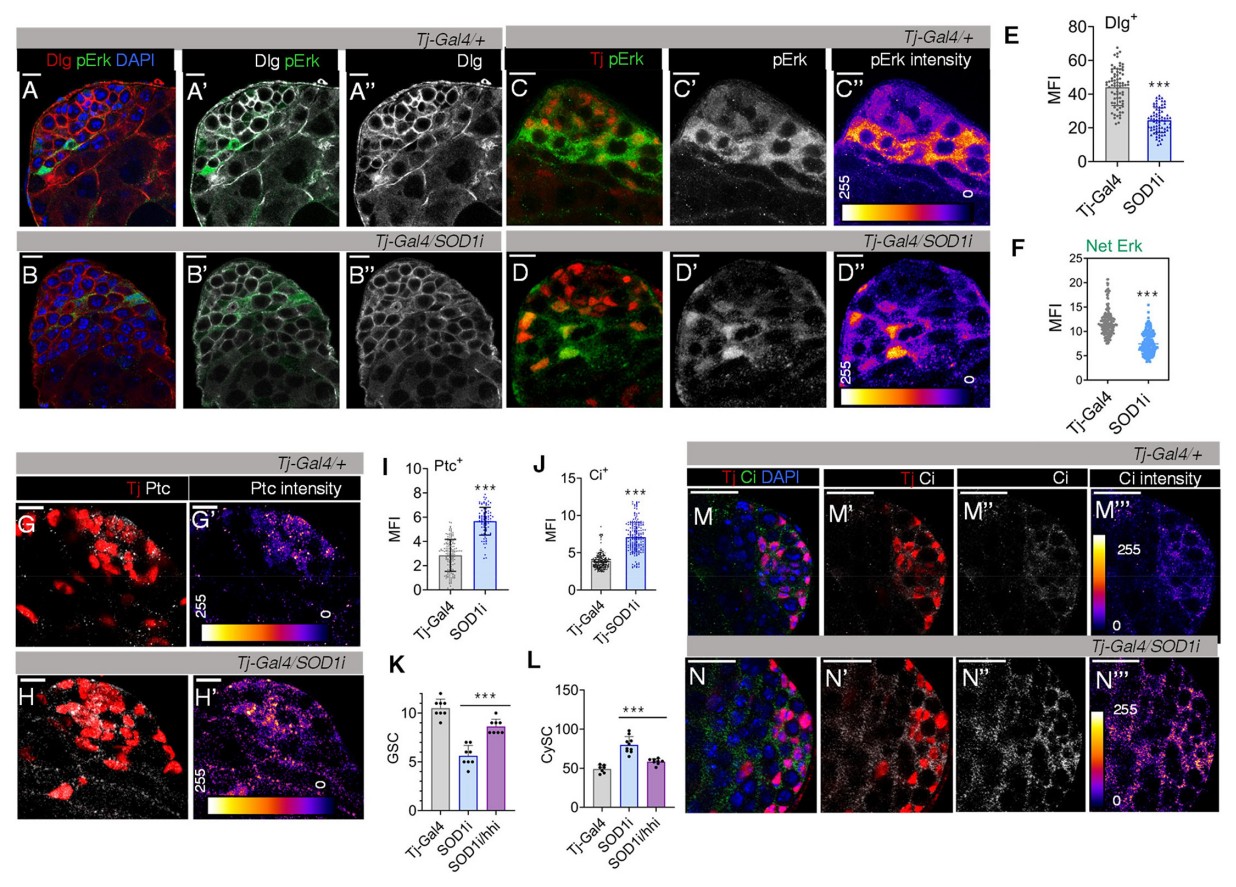

**Figure 4.** Altered cyst stem cells (CySC) redox state affects niche maintenance signals. (**A–B**) The expression of cell polarity marker discs-large (Dlg) (red) in control and *Sod1i* stem-cell niche (**A''-B''**), (**C–D**) Representative image showing pErk distribution parallel to Tj[+] cells and its expression pattern through Fire LUT (**C''-D''**). Scale bar - 10 µm. Mean fluorescence intensity (MFI) corresponding to Dlg level (**E**) was quantified, n=75, ***p (unpaired t-test)<0.0001. (**F**) MFI of total pErk expression was quantified, n=200, ***p (unpaired t-test)<0.0001. (**G'-H'**) Representative image showing distribution of Patched (Ptc) in Tj[+] cells in control and *Tj >Sod1* i. (**I–J**) Quantification of Ptc (**I**) and Hh effector Ci (**J**) expression through fluorescence intensity as mean ± s.e.m, n (Ptc)=90, n (Ci)=200, ***p (unpaired t-test)<0.0001 (**K–L**) The number of Vasa[+] (**K**) and Tj[+] (**L**) cells across control, *Sod1i*, and *Sod1i/Hhi* rescue samples depicted as mean ± s.e.m, n=10, ***p (unpaired t-test)<0.0001. Scale bar: 10 µm.

The online version of this article includes the following figure supplement(s) for figure 4:

**Figure supplement 1.** Activation of cyst stem cell (CySC) maintenance signalling promoting CySC proliferation.

## Elevating CySC antioxidant defence promotes GSC self-renewal

To further substantiate the role of ROS in coordinating the stem cell populations, we strengthened the cellular defences against oxidants by overexpressing Sod1 in CySCs. We checked the overall ROS level using DHE (*Figure 5—figure supplement 1I*) and monitored the relative GSC/CySC populations. Together with a reduction in superoxide levels, we observed an increase in the number of Vasa[+] cells (*Figure 5A*, *Figure 5—figure supplement 1A''-B''*), and a slight reduction in Tj[+] cells (*Figure 5B*, *Figure 5—figure supplement 1E''-F''*). Any morphological anomalies associated with cell-cell adhesion or abnormal cellular dispersion, was not observed (*Figure 5—figure supplement 1A–B and E-F*). Enhancing the levels of Sod1 in GSCs promoted the growth of GSC-like cells (*Figure 5—figure supplement 1C'', D'' and J*) but did not have any prominent effect on CySCs (*Figure 5—figure supplement 1G'', H'' and K*). The increment in Vasa[+] GSCs under CySC-induced low redox conditions was also represented by a parallel increase in spectrosome number (*Figure 5C–E*). The uptick in GSC number due to scavenging of ROS in CySC might be a result of delayed differentiation, as indicated by displacement of the Bam[+] zone away from the hub; thus, confirming the non-cell-autonomous role of CySC ROS in maintaining GSC fate (*Figure 5F–H*). The data suggest that reducing the redox profile

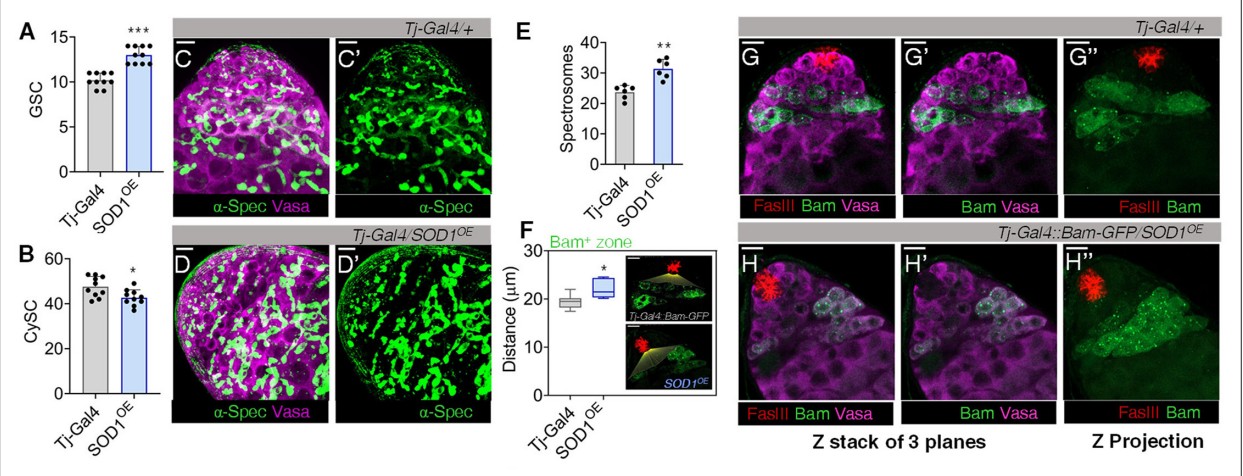

**Figure 5.** Low cyst stem cell (CySC) reactive oxygen species (ROS) sustains germline stem cell (GSC) maintenance. (**A–B**) Bar graphs illustrating variations in Vasa[+] GSC (**A**) and Tj[+] CySCs (**B**) numbers upon overexpressing Sod1 (Tj>Sod[OE]), represented as mean ± s.e.m, n=10, ***p (unpaired t-test)<0.0001, *p (unpaired t-test)<0.01. (**C–E**) The spectrosomes labelled with α-spectrin (α-spec) were imaged (**C'-D'**) and their number was quantified (**E**), n=10, **p (unpaired t-test)<0.001. (**F–H**) The initiation of gonialblast differentiation was determined by measuring the distance of Bam[+] zone from hub (red), n=10, *p (unpaired t-test)<0.01 (**F**), (**G''-H''**) shows distribution of Bam[+] reporter expressing cells. Scale bar: 10 µm.

The online version of this article includes the following figure supplement(s) for figure 5:

**Figure supplement 1.** Overexpression of Sod1 either in germline stem cells (GSCs) or cyst stem cells (CySCs) sustains self-renewing propensity of GSC-like cells.

of CySCs differentially affected their own self-renewing propensities along with GSC maintenance, and the optimum redox state of both stem-cell populations is controlled by CySCs.

## Discussion

ROS is a crucial mediator of stem cell maintenance where these species act as second messengers to induce different post-translational protein modifications, thereby affecting cell fate (*Sinenko et al., 2021*; *Liang and Ghaffari, 2014*). The effect of ROS is mainly considered to be autocrine in nature, where generation of these oxidative species is usually restricted to specific cellular compartments, ensuring they act close to their targets. Recent studies have reported non-autonomous generation of ROS by NOX enzymes or upon stimulation by growth factors to play a critical role in regenerative growth (*Fu et al., 2014*; *Taylor-Fishwick, 2013*; *Santabárbara-Ruiz et al., 2019*). However, in either case, the target cell serves as both the source and the site of response. Among ROS, hydrogen peroxide ($H_2O_2$) stands out due to its stability and membrane permeability (*Bienert et al., 2007*), allowing it to diffuse from its source (*Milkovic et al., 2019*). Non-myocytic pericardial cells (PCs) in *Drosophila* exhibit elevated ROS that act in a paracrine manner to influence adjacent cardiomyocytes (CMs) not by direct diffusion but through activation of the D-p38 MAPK cascade within PCs (*Lim et al., 2014*). In cardiomyocyte monolayers, wounding induces $H_2O_2$ and superoxide accumulation that propagate via gap junctions, creating a cell-to-cell ROS wave where even distant cells show increased cytosolic $H_2O_2$ and altered proteomes (*Fichman et al., 2024*). Together, these findings suggest that ROS can spread between cells either by direct diffusion/gap-junctional transfer or indirectly by modulating signaling cascades and secreted factors. Our findings indicate that CySCs, due to their high baseline ROS levels, affect the redox state of their neighbourhood, including GSCs. ROS imbalance in CySCs enhances the oxidative levels of their surrounding, that is, GSC that cause them to differentiate. High ROS-mediated differentiation of GSCs has already been reported earlier (*Tan et al., 2017*), and depletion of superoxide dismutases in GSCs did result in reduction of their number. However, perturbation of ROS in GSCs did not have a significant effect on CySC population. This shows that the cell non-autonomous effect of ROS is observed only when it originates from CySCs (*Figure 6*).

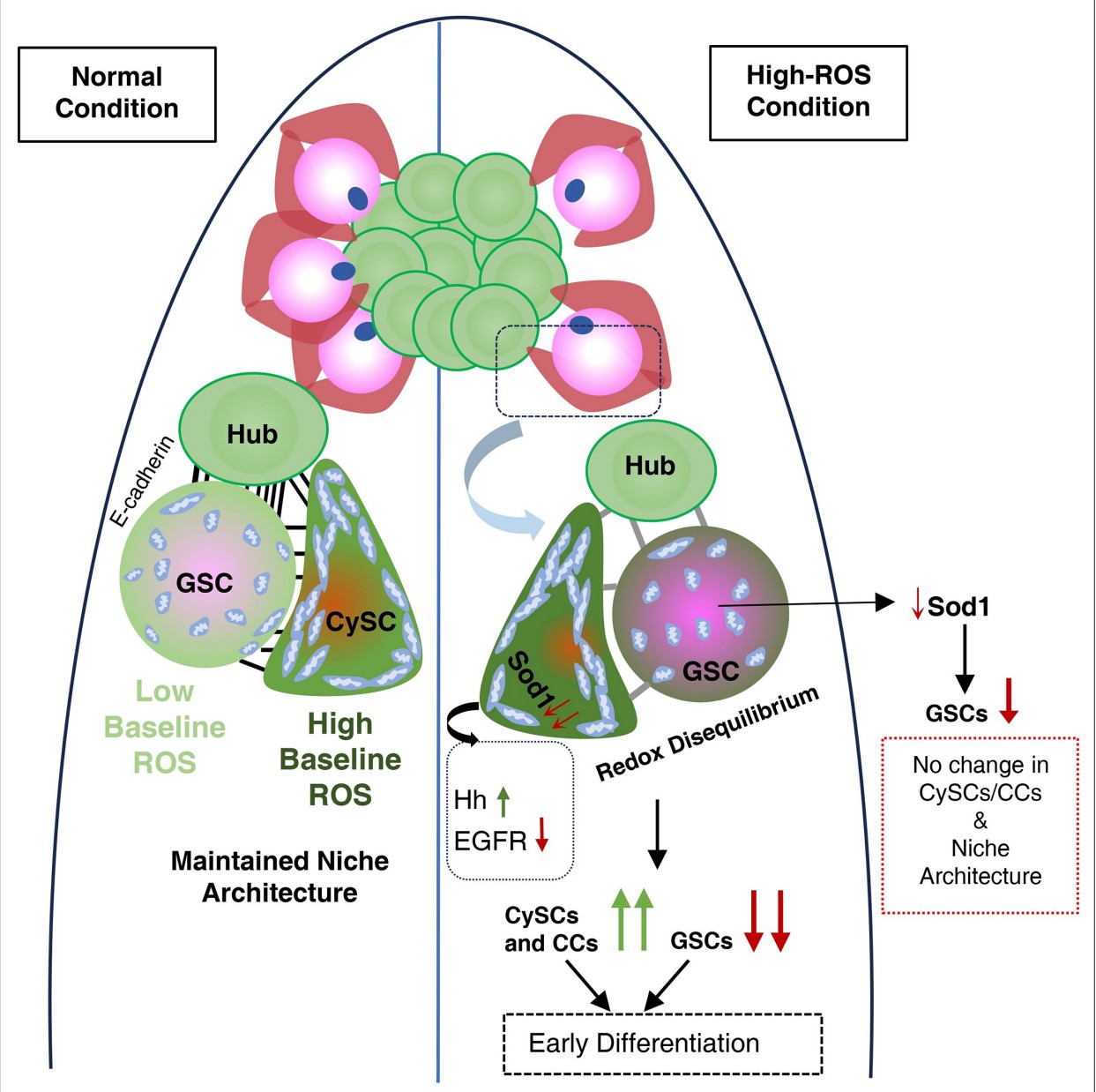

**Figure 6.** Proposed role of intercellular redox balance in germline maintenance and differentiation. germline stem cell (GSC) and cyst stem cell (CySC) are associated with different mitochondria abundance which corresponds to the presence of a redox differential between two stem cell populations. Disequilibrated redox potential among the two populations due to depletion of superoxide dismutase enhances CySC proliferation and precocious GSC differentiation, thereby disrupting the homeostatic balance between the two stem populations.

Our observations in *Drosophila* testicular stem cell niche suggested a previously unaccounted role of superoxide dismutases in maintaining niche homeostasis. The redox perturbations affected cyst stem cell dynamics, which indirectly influenced the germline. The higher ROS threshold of CySCs maintained the physiological redox state of GSCs. Reduced ROS in GSC was accompanied by its proliferation (*Pan et al., 2007*), phenocopied when Sod1 was overexpressed in its neighbours (*Figure 5*). However, we feel that net alterations in GSC state were the combinatorial effect of oxidative stress and niche alterations brought about by CySC deregulation.

GSCs are anchored to the hub cells via adherens junctions, allowing them to receive crucial maintenance signals (*Greenspan et al., 2015*). These include Bone Morphogenetic Proteins (BMPs), such as Decapentaplegic (Dpp) and Glass bottom boat (Gbb), which activate the BMP pathway (*Kawase*

*et al., 2004*) required for GSC self-renewal, and Unpaired (Upd), which triggers the JAK-STAT pathway to promote GSC adhesion to the hub (*Leatherman and Dinardo, 2010*). The suppression of these pathways activates Bam, acting as a switch from transit-amplifying cells to spermatocyte differentiation (*Bausek, 2013*; *Gönczy et al., 1997*). Elevated ROS in CySC possibly affected STAT phosphorylation through S-glutathionylation (*Butturini et al., 2014*; *Xiong et al., 2011*), reducing the levels of STAT-dependent transcripts, such as Socs36E (*Issigonis et al., 2009*), Ptp61F, and E-cadherin. This depletion compromises GSC adhesion, resulting in detachment from neighboring cells, similar to E-cadherin loss conditions (*Leatherman and Dinardo, 2010*; *DeGennaro et al., 2011*; *Karsten et al., 2002*; *Richmond et al., 2018*). This resulted in compromised GSC-CySC paracrine receptions, such as EGFR, which play a crucial role in the maintenance of cell polarity that segregates self-renewing CySC populations from the ones receiving differentiation cues (*Castanieto et al., 2014*; *Kiger et al., 2000*; *Schulz et al., 2002*), leading to the accumulation of cells co-expressing both stemness and differentiation markers. In contrast to GSCs, ROS imbalance in CySCs induced accelerated proliferation as well as differentiation due to deregulation of EGFR and Hedgehog pathways. Although redox modulation of EGFR pathway components has been previously demonstrated (*Koundouros and Poulogiannis, 2018*), in this study, we found Hedgehog to be probably susceptible to redox regulation. The redox-dependent induction of Hedgehog probably contributed to enhancement in CySC numbers, which partly contributed to depletion of GSCs.

However, we do not rule out the possibility of hub cells playing an equivalent role in GSC maintenance, given their parallel effect in suppressing GSC differentiation (*Tulina and Matunis, 2001*). Tj majorly expresses in CySCs but also shows modest expression in the hub. Therefore, the possible depletion of Sod1 in both hub and CySCs by *Tj-Gal4,* together with the demonstration of a stronger phenotype in *Tj-Sod1RNAi* testis compared to CySC-specific *C587-Sod1RNAi*, indicated potential hub contributions in CySC self-renewal. While ROS accumulation typically enhances EGFR signaling promoting premature GSC differentiation (*Sênos Demarco and Jones, 2019*), the loss of EGFR in somatic cells increases the number of GSCs (*Kiger et al., 2000*) and is associated with fewer somatic cells (*Amoyel et al., 2016a*). However, we observed that Sod1 depletion caused a parallel reduction of both pErk levels and GSC number. It is quite possible that the response of germline to Erk modulation is more context-dependent, which is influenced by other receptor tyrosine kinases beyond EGFR. To test this hypothesis, we had tried an EGFR gain-of-function rescue of CySC depletion, but the driven progenies were lethal in the absence of Sod1. We also avoided the usage of Gal80-dependent clonal populations to maintain a homogenous genetic background and prevent false readouts due to diffusion of ROS signals in unaltered neighbourhoods.

The proposed concept of intercellular ROS communication can be of importance in deciphering the biochemical adaptability and plasticity of different niches influencing stem cell fate, ensuring niche size and architecture to prevent stem cell loss and aging. This has been observed in neural stem cells where inflammation-induced quiescence recovers the regenerating capacity of aging brain (*Kalamakis et al., 2019*). In addition to metabolic variations, dependence of the germline on somatic neighbours for its redox state might be one of the reasons behind its presumed immortal nature and resistance to aging (*Snoeck, 2015*). The same can also be applied to cancer stem cells, which maintain niche occupancy and resist oxidative stress-induced apoptosis, probably by receiving redox cues from the environment. However, further work is required to elucidate the myriads of driver mechanisms intersecting in the realm of redox regulation that extend beyond the present system into broader translational areas.

## Materials and methods

**Key resources table**

| Reagent type (species) or resource | Designation | Source or reference | Identifiers | Additional information |
|---|---|---|---|---|
| Genetic reagent (*Drosophila melanogaster*) | *Oregon R+* | S.C. Lakhotia, BHU, India | | |
| Genetic reagent (*Drosophila melanogaster*) | *UAS-Sod1RNAi* | Bloomington *Drosophila* Stock Center | RRID:BDSC_24493 | RNAi present in Chr 2 |

*Continued on next page*

*Continued*

| Reagent type (species) or resource | Designation | Source or reference | Identifiers | Additional information |
|---|---|---|---|---|
| Genetic reagent (*Drosophila melanogaster*) | *UAS-Sod1RNAi* | Bloomington *Drosophila* Stock Center | RRID:BDSC_32909 | RNAi present in Chr 3 |
| Genetic reagent (*Drosophila melanogaster*) | *UAS-Sod1RNAi* | Bloomington *Drosophila* Stock Center | RRID:BDSC_29389 | RNAi present in Chr 3 |
| Genetic reagent (*Drosophila melanogaster*) | *UAS-Sod2RNAi* | Bloomington *Drosophila* Stock Center | RRID:BDSC_32983 | |
| Genetic reagent (*Drosophila melanogaster*) | *UAS-hhRNAi* | Bloomington *Drosophila* Stock Center | RRID:BDSC_32489 | |
| Genetic reagent (*Drosophila melanogaster*) | *Nos-Gal4* | Bloomington *Drosophila* Stock Center | RRID:BDSC_25751 | |
| Genetic reagent (*Drosophila melanogaster*) | *UAS-FUCCI* | Bloomington *Drosophila* Stock Center | RRID:BDSC_55122 | |
| Genetic reagent (*Drosophila melanogaster*) | *Tj-GAL4* | Pralay Majumder, Presidency University, India | | |
| Genetic reagent (*Drosophila melanogaster*) | *Tj-Gal4 bamGFP/Cyo* | Krishanu Ray, TIFR-Mumbai, India | | |
| Genetic reagent (*Drosophila melanogaster*) | *gstD1-GFP* | B.C. Mandal, BHU | | |
| Genetic reagent (*Drosophila melanogaster*) | *TFAM-GFP* | Hong Xu, NIH, USA | | |
| Antibody | Anti-FasIII (Mouse monoclonal) | Developmental Studies Hybridoma Bank (DSHB) | Cat# 7G10, RRID:AB_528238 | IF (1:120) |
| Antibody | Anti-Eya (Mouse monoclonal) | DSHB | Cat# eya10H6, RRID:AB_528232 | IF (1:20) |
| Antibody | Anti DE-Cadherin (Rat monoclonal) | DSHB | Cat# DCAD2, RRID:AB_528120 | IF (1:20) |
| Antibody | Anti-α-Spectrin (Mouse monoclonal) | DSHB | Cat# 3A9, RRID:AB_528473 | IF (1:50) |
| Antibody | Anti beta-tubulin (Mouse monoclonal) | DSHB | Cat# E7, RRID:AB_528499 | Western (1:300) |
| Antibody | Anti-Phospho-p44/42 MAPK (Rabbit monoclonal) | Cell Signaling Technology | Cat# 4370, RRID:AB_2315112 | IF (1:100) |
| Antibody | anti-ATP5A 915H4C4 (Mouse monoclonal) | Abcam | Cat# ab14748, RRID:AB_301447 | IF (1:700) |
| Antibody | Anti-Traffic jam (Tj) | Godt (University of Toronto, Canada) | | IF (1:5000) |
| Antibody | Anti-Vasa | Lehmann (Whitehead Institute, USA) | | IF (1:4000) |
| Antibody | Anti-Zfh1 | Lehmann (Whitehead Institute, USA) | | IF (1:2000) |
| Sequence-based reagent | CycD (F) | This paper | PCR primers | CCAGAACAATGCCGTAGTGTG |
| Sequence-based reagent | CycD (R) | This paper | PCR primers | AACGCGGATAACTTTGGATTGA |
| Sequence-based reagent | Hh (F) | This paper | PCR primers | CGCCAGTGTCACCTGTCTC |
| Sequence-based reagent | Hh (R) | This paper | PCR primers | TTCTTGCGGGATTGCGGAG |
| Sequence-based reagent | Ptc (F) | This paper | PCR primers | TTCCAGTCCCACCTCGAAAC |
| Sequence-based reagent | Ptc (R) | This paper | PCR primers | GATCGTCTTCTGTGTGTAGGC |

*Continued*

| Reagent type (species) or resource | Designation | Source or reference | Identifiers | Additional information |
|---|---|---|---|---|
| Sequence-based reagent | Smo (F) | This paper | PCR primers | CTGTTTCGGCTCAAAATTGCC |
| Sequence-based reagent | Smo (R) | This paper | PCR primers | GTAGTCGTTCAGCTTATCGTTCA |
| Sequence-based reagent | Ci (F) | This paper | PCR primers | GATTTTCGCCAAACTCTTTAGCC |
| Sequence-based reagent | Ci (R) | This paper | PCR primers | ACATGGGATTAAGGGCGGTAG |
| Sequence-based reagent | RP49 (F) | This paper | PCR primers | TTCAAGATGACCATCCGC |
| Sequence-based reagent | RP49 (R) | This paper | PCR primers | TTAGCATATCGATCCGACTG |
| commercial assay or kit | Rneasy Mini kit | Qiagen | Cat# 74104 | |
| Chemical compound, drug | DAPI stain | Invitrogen | D1306 | (1 µg/mL) |
| Chemical compound, drug | Phosphatase inhibitor cocktail 2 | Sigma | Cat# P5726 | (1:100) |
| Chemical compound, drug | PIPES | Sigma | Cat# P6757 | |
| Chemical compound, drug | EGTA | Sigma | Cat# E4378 | |
| Software, algorithm | ImageJ | NIH | RRID:SCR_003070 | https://imagej.nih.gov/ij/download.html |
| Software, algorithm | GraphPad Prism | GraphPad Software | RRID:SCR_002798 | Version 8 |
| Software, algorithm | Adobe Photoshop 2021 | Adobe | RRID:SCR_014199 | Version 22.4.2 |

## Fly strains

Fly stocks were maintained and crosses were set at 25 °C on normal corn meal and yeast medium unless otherwise indicated. All fly stocks (BL-24493) *UAS SOD1 RNAi*, (BL-32909) *UAS SOD1 RNAi*, (BL-29389) *UAS SOD1 RNAi*, (BL-32983) *UAS SOD2 RNAi*, (BL-24754) *UAS SOD1*, (BL-32489) *UAS hhRNAi*, (BL-25751) *UAS Dcr-nos Gal4*, (BL-55122) *UAS-FUCCI*, were obtained from Bloomington *Drosophila* Stock Centre. *Nos-Gal4*, *Tj-Gal4*, *Tj-Gal4-BamGFP*, *gstD1-GFP*, *TFAM-GFP*, *UAS-Ci* were kind gifts from U. Nongthomba, P. Majumder, K. Ray, B. C. Mandal, Hong Xu, LS Shahidhara labs, respectively. Parents were maintained at 25 °C, and crosses were set at the same temperature. To ensure optimal Gal4 activity, progenies were shifted to 29 °C until eclosion, with aging also conducted at 29 °C. Males aged 3–5 days were used for all experiments. The control lines for all experiments are the corresponding Gal4 line crossed with wild-type *OregonR⁺*.

## Dissection and immunostaining

Anesthetised flies were dissected in 1 X Phosphate Buffer Saline (PBS). All incubations were carried out at room temperature (25 °C) unless otherwise mentioned. Testes were fixed in 4% paraformaldehyde for 30 min, followed by three washes with 0.3% PBTX (PBS+TritonX 100). Post-blocking in 0.5% Bovine Serum Albumin, testis was incubated overnight at 4 °C for primary antisera, followed by washing in 0.3% PBTX three times 15 min each before incubating with secondary antibody. The tissues were counterstained with DAPI (4,6-diamidino-2-phenylindole) for 20 min, followed by three washes in 0.3% PBTX for 15 min each. For Patched antibody staining, a modified protocol utilizing PIPES-EGTA buffer was used as described (*Michel et al., 2011*). For Eya staining, the tissues were incubated in primary antibody for 2 days. The dpErk was labelled by dissecting fly testes in Schneider's media, followed by fixation in 4% PFA for 30 min, 3 x wash in 0.1% PBTX, blocked and incubated in primary antibody overnight; each step supplemented with phosphatase inhibitor cocktail 2 (1:100, Sigma, cat#P5726). Samples were mounted in DABCO anti-fade medium prior to imaging.

The following primary antibodies were used in the experiments - anti-FasIII (7G10) (1:120, DSHB (Developmental Studies Hybridoma Bank)), anti-Eya (1:50, DSHB), anti-DEcad (DCAD2) (1:50, DSHB), anti-α-Spectrin (3A9) (1:20, DSHB), anti-Ptc (1:50, DSHB), anti-Ci (1:50, DSHB), anti-Dlg (1:50), anti-β tubulin (1:300, DSHB), anti-pERK (1:100, Cell Signalling (4370)), ATP5A (1:700, Abcam (ab14748)),

anti-Tj (1:5000), anti-Vasa (1:4000), anti-Zfh1 (1:2000). The primary labelling was detected using appropriate Alexa-Fluor tagged secondary antibody (Thermo Fisher Scientific).

To assess superoxide levels, testes were dissected in Schneider's medium (SM) and immediately incubated in 30 µM Dihydroxyethidium (DHE) at 25 °C in the dark. Testes were washed thrice with SM for 7 min each before quantifying the emitted fluorescence using SpectraMax iD5 (Molecular Devices).

## Quantitative reverse transcription–PCR

For semi-quantitative RT-PCR and qRT-PCR analyses, total RNA was isolated from testes of 3–5 days old male flies using Qiagen RNA extraction kit. RNA pellets were resuspended in nuclease-free water, and the quantity of RNA was spectrophotometrically estimated. First-strand cDNA was synthesised from 1 to 2 µg of total RNA as described earlier. The prepared cDNAs were subjected to real-time PCR using forward and reverse primer pairs as listed below, using 5 µl qPCR Master Mix (SYBR Green, Thermo Fisher Scientific), 2 pmol/µl of each primer per reaction for 10 µl final volume in ABI 7500 Real-time PCR machine. The fold change in expression was calculated through $2^{-\Delta\Delta Ct}$ methods. The primers used are listed in the Key Resources Table.

## Imaging and image analysis

Confocal imaging was carried out using Zeiss LSM 900 confocal microscope, using appropriate dichroics and filters. Images were further analysed and processed for brightness and contrast adjustments using ImageJ (Fiji). Mean intensity measurements were carried out using standard Fiji plug-ins. The relative distance from the two reference points in the images was estimated through Inter-edge Distance ImageJ Macro v2.0 from Github. All images were assembled using Adobe Photoshop version 21.2.1.

## Quantification of GSCs, CySCs, and CCs

For GSC quantification, only cells in direct contact with the hub were considered. While single slices are shown for representation, counting was performed on individual slices of z-stacks using the semi-automated Cell Counter plugin in Fiji for all specified genotypes.

For CySC quantification, all Tj-positive and Zfh1-positive cells present in each testis were counted and represented. For cyst cell quantification, only those near the hub were considered, which does not reflect the total number of Eya-positive cells. All quantifications were performed using the Cell Counter plugin in Fiji.

## Mitochondrial and *gstD1*-GFP quantification

The mitochondrial analyzer first generates 2D mitochondrial regions of interest (ROIs), followed by the measurement of their total area and circularity. Initially, this is done using a thresholded image of a single slice. For *gstD1*-GFP quantification, we have quantified the region of *gstD1*-GFP that overlapped with Vasa-positive germ cells which are in direct contact with the hub to assess the presence of ROS reporter signal within the germline compartment. To ensure accurate cell boundary demarcation, we used Dlg staining as an additional parameter.

## Immunoblot analysis

Protein was extracted from the dissected testes using RIPA buffer as described previously and quantified using Bradford reagent (Biorad). Equivalent concentration of lysate was denatured using 1 X Laemmli Buffer with 1 M DTT at 95 °C, separated in 10% SDS-PAGE, and transferred onto PVDF membranes. Membranes were blocked with commercial blocking solution in TBST base before sequential primary antibody incubation with anti-Vasa and anti-β-Tubulin (DSHB, E7). Secondary detection was performed using HRP-tag anti-mouse antibody (GE Amersham).

## Statistical analyses

The sample sizes carrying adequate statistical power are mentioned in the Figure legends. Statistical significance for each experiment was calculated using two-tailed Student's t-test, unless otherwise mentioned, using GraphPad Prism 8 software. The $p$-values were calculated through pairwise comparison of the data with wild-type or driver alone and driven RNAi lines. Significance values for sample sizes mentioned in Figure legends were represented as $*p<0.01$, $**p<0.001$, or $***p<0.0001$.

## Acknowledgements

We thank P Majumder, U Nongthomba, K Ray, G Ratnaparkhi, BC Mandal, Hong Xu lab, and SC Lakhotia for kindly sharing some of the transgenic fly lines. Christian Bökel for sharing experimental protocols. Anti-Tj antibody was a generous gift from Prof. D Godt, University of Toronto. Anti-Vasa and Anti-Zfh1 were kindly gifted by Prof. R Lehmann, New York University. We also thank Dr. Bama Charan Mandal for generously sharing his confocal microscope imaging system and Sudeshna Majumder for preliminary data. We acknowledge BDSC for fly stocks and DSHB for antibodies. This work is supported by the Innovative Young Biotechnologist Award by the Department of Biotechnology (BT/12/IYBA/2019/01), Early Career Research Award by the Science and Engineering Research Board (ECR/2018/000009), BHU-Institute of Eminence grant (R/Dev/IoE/Incentive/2021-22/32452). The authors also acknowledge financial support from the DST-INSPIRE fellowship (to OM) and the UGC-CAS doctoral fellowship (to TC).

## Additional information

### Funding

| Funder | Grant reference number | Author |
|---|---|---|
| Department of Biotechnology, Ministry of Science and Technology, India | BT/12/IYBA/2019/01 | Devanjan Sinha |
| Science and Engineering Research Board | ECR/2018/000009 | Devanjan Sinha |
| BHU-Institute of Eminence grant | R/Dev/IoE/Incentive/2021-22/32452 | Devanjan Sinha |
| DST-INSPIRE fellowship | | Olivia Majhi |
| UGC-CAS doctoral fellowship | | Tanvi Chaudhary |

The funders had no role in study design, data collection and interpretation, or the decision to submit the work for publication.

### Author contributions

Olivia Majhi, Conceptualization, Resources, Data curation, Software, Formal analysis, Methodology, Writing – original draft, Writing – review and editing; Aishwarya Chhatre, Tanvi Chaudhary, Investigation, Methodology; Devanjan Sinha, Conceptualization, Resources, Data curation, Formal analysis, Supervision, Funding acquisition, Project administration, Writing – review and editing

### Author ORCIDs

Olivia Majhi ⓘ https://orcid.org/0009-0000-7898-014X
Aishwarya Chhatre ⓘ https://orcid.org/0009-0008-4243-9093
Tanvi Chaudhary ⓘ https://orcid.org/0009-0002-8244-4395
Devanjan Sinha ⓘ https://orcid.org/0000-0001-5060-2075

Reviewer #1 (Public review): https://doi.org/10.7554/eLife.96446.4.sa1
Author response https://doi.org/10.7554/eLife.96446.4.sa2

## Additional files

### Supplementary files

MDAR checklist

Source data 1. Raw data for graphs.

## Data availability

The data that support the findings of this study are available within the main text and its Source data file. All information required for data reproducibility, including information on key reagents, protocols, etc are included in the main text. This study did not utilize or generate any unique datasets or codes.

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
