## [Editor Report · eLife Assessment]

In this work, the authors intend to assess the existence of a redox potential across germline stem cells and neighbouring somatic stem cells in the Drosophila testis. Some aspects of the manuscript are **convincing**, like the clear effect of SOD KD on cyst cell differentiation state. Other conclusions of the work, such as the non-autonomous effect of this KD on germ cells are not sufficiently supported by the data. This remains true even with the revised version of the paper, as the effect of redox state of the soma on the germline is a major point of the paper, and this remains a critical flaw. The work could be potentially **useful** if the critiques of the reviewers were fully addressed; the strength of the evidence of the manuscript as it stands is still **inadequate**. Readers should use their own judgment about the validity and meaningfulness of different findings.

---

## [Referee Report · Reviewer #1 (Public review)]

Mitochondrial staining difference is convincing, but the status of the mitos, fused vs fragmented, elongated vs spherical, does not seem convincing. Given the density of mito staining in CySC, it is difficult to tell what is an elongated or fused mito vs the overlap of several smaller mitos.

I'm afraid the quantification and conclusions about the gstD1 staining in CySC vs. GSCs is just not convincing-I cannot see how they were able to distinguish the relevant signals to quantify once cell type vs the other.

The overall increase in gstD1 staining with the CySC SOD KD looks nice, but again I can't distinguish different cel types. This experiment would have been more convincing if the SOD KD was mosaic, so that individual samples would show changes in only some of the cells. Still, it seems that KD of SOD in the CySC does have an effect on the germline, which is interesting.

The effect of SOD KD on the number of less differentiated somatic cells seems clear. However, the effect on the germline is less clear and is somewhat confusing. Normally, a tumor of CySC or less differentiated Cyst cells, such as with activated JAK/STAT, also leads to a large increase in undifferentiated germ cells, not a decrease in germline as they conclude they observe here. The images do not appear to show reduced number of GSCs, but if they counted GSCs at the niche, then that is the correct way to do it, but its odd that they chose images that do not show the phenotype. In addition, lower number of GSCs could also be caused by "too many CySCs" which can kick out GSCs from the niche, rather than any affect on GSC redox state. Further, their conclusion of reduced germline overall, e.g. by vasa staining, does not appear to be true in the images they present and their indication that lower vasa equals fewer GSCs is invalid since all the early germline expresses Vasa.

The effect of somatic SOD KD is perhaps most striking in the observation of Eya+ cyst cells closer to the niche. The combination of increased Zfh1+ cells with many also being Eya+ demonstrates a strong effect on cyst cell differentiation, but one that is also confusing because they observe increases in both early cyst cells (Zfh1+) as well as late cyst cells (Eya+) or perhaps just an increase in the Zfh1/Eya double-positive state that is not normally common. The effects on the RTK and Hh pathways may also reflect this disturbed state of the Cyst cells.

However, the effect on germline differentiation is less clear-the images shown do not really demonstrate any change in BAM expression that I can tell, which is even more confusing given the clear effect on cyst cell differentiation.

For the last figure, any effect of SOD OE in the germline on the germline itself is apparently very subtle and is within the range observed between different "wt" genetic backgrounds.

Comments on revisions:

Upon re-re-review, the manuscript is improved but retains many of the flaws outlined in the first reviews.

---

## [Author Response]

The following is the authors’ response to the previous reviews.

**Public Reviews:**

**Reviewer #1 (Public review)**
Mitochondrial staining difference is convincing, but the status of the mitochondria, fused vs fragmented, elongated vs spherical, does not seem convincing. Given the density of mito staining in CySC, it is difficult to tell whether what is an elongated or fused mito vs the overlap of several smaller mitos.

To address this, we have now removed the statements regarding the differences in the shape of mitochondria among the stem cell population. We have limited our statements to stating that the CySCs are more mitochondria dense compared to the neighbouring GSCs.

The quantification and conclusions about the gstD1 staining in CySC vs. GSCs is just not convincing-I cannot see how they were able to distinguish the relevant signals to quantify once cell type vs the other.

We appreciate the reviewer’s concern. To address this, we have included new images along with z-stack reconstructions (Fig 1G-P and S1C-D’’’), which now provide clearer distinction of gstD1 staining between CySCs and GSCs and improve the accuracy of quantification. The intensity of gstD1 staining overlapping with that of Vasa+ zone has been quantified as ROS levels for GSCs. Similarly, the cytoplasmic area of gstD1 stain bounded by Dlg and Tj+ nuclei was quantified as ROS levels for CySCs.

Images do not appear to show reduced number of GSCs, but if they counted GSCs at the niche, then that is the correct way to do it, but its odd that they chose images that do not show the phenotype. Further, their conclusion of reduced germline overall, e.g by vasa staining, does not appear to be true in the images they present and their indication that lower vasa equals fewer GSCs is invalid since all the early germline expresses Vasa.

We have replaced the figure with images where the GSC rosette is clearly visible, ensuring that the counted GSCs at the niche accurately reflect the phenotype (Fig. 2 C’’, D’’). We agree that Vasa is expressed in all early germline cells. The overall reduced Vasa signal intensity in our western blot analysis for Sod1RNAi reflects a general reduction in the germline population, not just the GSCs. We have modified our statements in the Results appropriately.

However, the effect on germline differentiation is less clear-the images shown do not really demonstrate any change in BAM expression that I can tell, which is even more confusing given the clear effect on cyst cell differentiation.

We appreciate the reviewer’s observation. To clarify this point, we have now included z-stack projection images of Bam expression in the revised version (Fig 3E’’-F’’) .

These images more clearly demonstrate the difference in Bam expression, thereby highlighting the effect on germline differentiation. Moreover, Bam expressing cells are present more closure to hub in Sod1RNAi condition, indicating early differentiation.

For the last figure, any effect of SOD OE in the germline on the germline itself is apparently very subtle and is within the range observed between different "wt" genetic backgrounds.

We acknowledge that the effect of SOD overexpression on the germline is not very significant. The germline cells already possess a modest ROS load and it is a well-established fact that they possess a robust anti-oxidant defence machinery in order to protect the genome. Therefore, elevating the levels of antioxidant enzymes such as Sod1 does not translate into a major change and the effect observed are generally subtle.

**Reviewer #3 (Public review)**
In Fig. 1N (tj-SODi), one can see that all of gst-GFP resides within the differentiating somatic cells and none is in the germ cells. Furthermore, the information provided in the materials and methods about quantification of gst-GFP is not sufficient. Focusing on Dlg staining is not sufficient. They need to quantify the overlap of Vasa (a cytoplasmic protein in GSCs) with GFP.

In our analysis, we have indeed quantified the GFP intensity in area of overlap between gstD1-GFP and Vasa-positive zone in the germ cells which are in direct contact with hub, in order to accurately quantify the ROS reporter signal within the germline compartment. Further, to ensure accurate cell boundary demarcation, we used Dlg staining as an additional parameter. While Dlg staining alone was included in the figure panels for clarity of visualization, the actual quantification was performed by considering both Vasa (for germ cells cytoplasm) and Dlg (for cellular boundaries). This has been clarified in the Materials and Methods.

Additionally, since Tj-gal4 is active in hub cells, it is not clear whether the effects of SOD depletion also arise from perturbation of niche cells.

We acknowledge that Tj-Gal4 also shows minimal activity in hub cells. To address this, we had tested C587-Gal4 and observed similar effects on niche architecture, though weaker than with Tj-Gal4, underlying the effect of ROS originating from CySC.

First, the authors are studying a developmental effect, rather than an adult phenotype. Second, the characterization of the somatic lineage is incomplete. It appears that high ROS in the somatic lineage autonomously decreases MAP kinase signaling and increases Hh signaling. They assume that the MAPK signaling is due to changes in Egfr activity but there are other tyrosine kinases active in CySCs, including PVR/VEGFR (PMID: 36400422), that impinge on MAPK. In any event ,their results are puzzling because lower Egfr should reduce CySC self-renewal and CySC number (Amoyel, 2016) and the ability of cyst cells to encapsulate gonialblasts (Lenhart Dev Cell 2015). The increased Hh should increase CySC number and the ability of CySCs to outcompete GSCs. The fact that the average total number of GSCs declines in tj>SODi testes suggests that high ROS CySCs are indeed outcompeting GSCs. However, as I wrote in myfirst critique, the characterization of the high ROS soma is incomplete. And the role of high ROS in the hub cells is acknowledged but not investigated.

We acknowledge the reviewer’s concern that our study primarily examines a developmental effect. Our rationale was that redox imbalance during early stages can set longterm trajectories for stem cell behavior and niche organization, which ultimately manifest in adult testes.

We agree that sole evaluation of Erk levels may not reflect the actual status of EGFR signalling and there is an apparent contradictory observation of low Erk and high CySC self-renewal. We believe that this ROS mediated change in Erk status, resulting in high CySC proliferation, might be an outcome of an interplay between other RTKs beyond EGFR. While the expansion of CySCs is primarily governed by Hh, a detailed dissection of these pathways under altered redox environment will be an interesting work to develop in future. Regarding the GSC number, it cannot be definitively stated that high ROS-CySCs are indeed outcompeting the GSCs, but yes, that possibility parallely exists. However, in presence case, there is no denying that the ROS levels of GSCs are indeed high under high CySC-ROS condition. It is known that ROS imbalance in GSCs promote their differentiation which was also observed in the present study through Bam staining. Therefore, redox mediated reduction in GSC number cannot be completely ruled out. We have already discussed these points in the revised manuscript and suggest possible non-canonical effects of ROS on signal integration within CySCs that might reconcile these findings. Further, in the present study, we have focussed on redox interplay between the two stem cell populations (GSC and CySC) of the niche. Hence, we have not covered the redox profiling of the hub in detail.

The paragraph in the introduction (lines 62-76) mentions autonomous ROS levels in stem cells, not the transfer of ROS from one cell to another. And this paragraph is confusing because it starts with the (inaccurate) statement all stem cells have low ROS and then they discuss ISCs, which have high ROS.

We have revised the paragraph for clarity. It now distinguishes between stem cell types with low versus relatively high ROS requirements (e.g., ISCs, HSCs, NSCs) and includes recent evidence of non-autonomous ROS signaling, such as paracrine ROS action from pericardial cells to cardiomyocytes and gap-junction–mediated ROS waves in cardiomyocyte monolayers. This resolves the ambiguity and presents a balanced view of autonomous and nonautonomous ROS regulation.

While there has been an improvement in the scholarship of the testis, there are still places where the correct paper is not cited and issues with the text.

All concerns regarding missing or incorrect citations and textual issues have now been carefully addressed and corrected. Relevant references have been added in the appropriate places to ensure accuracy.

The authors are encouraged to more completely characterize the phenotype of high ROS in hub and CySCs.

We have now included improved images showing the respective ROS profiles GSCs, CySCs and the hub. As mentioned in the earlier response, this work focuses on the redox interplay between GSCs and CySCs hence, we have not included any analysis on hub. However, we agree with reviewer that the hub contributions should also be evaluated as a future direction.